# Creatine Improves Total Sleep Duration Following Resistance Training Days versus Non-Resistance Training Days among Naturally Menstruating Females

**DOI:** 10.3390/nu16162772

**Published:** 2024-08-20

**Authors:** Ariel J. Aguiar Bonfim Cruz, Samantha J. Brooks, Katelyn Kleinkopf, C. J. Brush, Gena L. Irwin, Malayna G. Schwartz, Darren G. Candow, Ann F. Brown

**Affiliations:** 1Department of Movement Sciences, College of Education, Health & Human Sciences, University of Idaho, Moscow, ID 83844, USA; ariela@uidaho.edu (A.J.A.B.C.); sbrooks.phd@gmail.com (S.J.B.); klei4180@vandals.uidaho.edu (K.K.); cbrush@uidaho.edu (C.J.B.); girwin@uidaho.edu (G.L.I.); 2WWAMI Medical Education Program, University of Idaho, Moscow, ID 83844, USA; mhambly@uw.edu; 3Aging Muscle & Bone Laboratory, Faculty of Kinesiology & Healthy Studies, University of Regina, Regina, SK S4S 0A2, Canada; darren.candow@uregina.ca

**Keywords:** creatine monohydrate, sleep, resistance exercise, female health, body composition

## Abstract

Females historically experience sleep disturbances and overall poor sleep compared to males. Creatine has been proposed to impact sleep; however, the effects are not well known. The purpose of this study was to examine the effects of creatine supplementation on sleep among naturally menstruating females. Twenty-one participants completed a double-blind, randomized controlled trial in which they consumed 5 g creatine + 5 g maltodextrin or placebo, 10 g maltodextrin, daily for 6 weeks. Participants completed resistance training 2x/week using the TONAL^®^ (Tonal Systems Inc., San Francisco, CA, USA) at-home gym. Pre- and post-testing assessed body composition, Pittsburgh Sleep Quality Index (PSQI), dietary intake, and muscular strength. Sleep was assessed nightly using an ŌURA^®^ (Oulu, Finland) ring. Compared to the placebo group, those consuming creatine experienced significant increases in total sleep on training days (*p* = 0.013). No significant changes in chronic sleep and PSQI (pre–post) were observed. There was a significant increase in TONAL^®^ strength score over time (*p* < 0.001), with no between-group differences. Participants reduced their total calorie (kcal) (*p* = 0.039), protein (g/kg) (*p* = 0.009), carbohydrate (g/kg) (*p* = 0.023), and fat (g) (*p* = 0.036) intake over time. Creatine supplementation increases sleep duration on resistance training days in naturally menstruating females.

## 1. Introduction

Creatine (α-methyl guandino-acetic acid; CR) is endogenously synthesized in the liver and brain from reactions involving the amino acids arginine, glycine, and methionine. Exogenously, CR can be consumed in the diet (primarily from red meat and seafood) or through commercially manufactured creatine products [1]. On average, ≤5% of total body creatine stores are located in the brain [2]. CR supplementation has the ability (albeit in limited capacity) to cross the blood–brain barrier and increase brain creatine levels [3]. Using magnetic resonance spectroscopy, there is accumulating research showing that CR supplementation has favorable effects on indices of brain health and function [3,4,5,6]. Specifically, CR supplementation may affect cognitive processes related to sleep deprivation, which could have important implications on sleep quality, continuity, and quantity [3,7,8].

Despite multiple physiological mechanisms to replenish CR stores, female storage is ~70–80% lower compared to males [9]. Further, females often consume less dietary CR compared to males, likely due to lower meat consumption [9]. Thus, CR supplementation may be particularly useful given lower endogenous stores and/or exogenous consumption of CR which may contribute to overall female health. Young adult females experience greater severity of sleep disturbances and overall worse sleep quality compared to their male counterparts [10]. These findings could be attributed to menstrual cycle hormone fluctuations’ impact on circadian rhythm [11,12]. Baker and Driver identified that sleep quality was impacted by different phases of the menstrual cycle, but not sleep continuity [13]. Further, disrupted sleep has been associated with altered menstrual cycle phases [14]. Yet, Alzueta et al. examined sleep throughout the menstrual cycle measured by the ŌURA^®^ ring and identified that sleep did not significantly vary by menstrual cycle phase [15]. Although sleep is likely altered throughout the menstrual cycle, it remains unclear if CR influences sleep quality or quantity in the female population.

The positive effects of resistance training on muscular strength and body composition (e.g., lean soft tissue (LST), fat mass (FM)) are well established [5,16]. These beneficial effects are further augmented with CR supplementation [17,18]. Resistance training also improves sleep quality and quantity [16]. Despite its widely accepted benefits, only 24.3% of females in 2018 met the resistance training recommendations outlined in the Physical Activity Guidelines for Americans, 2nd ed. 2018 [19]. Given the beneficial effects of resistance training and CR supplementation, research is warranted specifically among females due to menstrual-cycle-mediated changes in circadian rhythm. Therefore, the primary purpose of this study was to examine the effects of CR supplementation and resistance training on sleep among naturally menstruating pre-menopausal females. A secondary purpose was to examine the effects of CR supplementation on measures of body composition and muscle strength. It was hypothesized that CR supplementation and resistance training would improve sleep quality more than resistance training alone. Secondarily, it was hypothesized that CR supplementation would augment the gains in LST and strength and decrease FM more than those on placebo (PLA).

## 2. Materials and Methods

### 2.1. Participants

Twenty-seven naturally menstruating, non-resistance-trained (≤1 day/week) females (18–24 years of age) were enrolled in this study. An a priori power analysis (G*Power v. 3.1.9.7; [20]) indicated that 24 participants were required to reach a moderate effect size (Cohen’s *d* = 0.5), alpha level of 0.05, and power of 0.80 for repeated measures design with two groups. Participants were included if they were naturally menstruating or using a copper intrauterine device (IUD). Participants were excluded if they were not biologically female, using any type of hormonal birth control, amenorrheic, regularly participating in resistance training (>1 day/week), consuming CR supplementation, vegan or vegetarian, using medications that could affect muscle biology (i.e., corticosteroids), if they had any pre-existing liver or kidney abnormalities, or if the responded ‘yes’ to any questions on the Physical Activity Readiness Questionnaire for Everyone (PARQ+). Participants were instructed not to change their habitual diet or engage in planned resistance training outside of the intervention training program. This study was approved by the Institutional Review Board at the University of Idaho and registered as a clinical trial (NCT05745870).

### 2.2. Experimental Design

This study was a randomized, double-blind, placebo-controlled, 6-week intervention. To minimize differences between groups at baseline, participants were matched based on body weight (kg), current physical activity frequency (moderate and vigorous minutes/week) and habitual animal protein consumption (g). Participants were then randomized into either the CR group or PLA group. All participants were enrolled in the resistance training program to standardize exercise throughout the study. Once randomized, participants began supplementation and resistance training concurrently (Figure 1). Menstrual cycle phase was noted at the beginning of the intervention.

The primary outcome variables measured daily included sleep quantity (ŌURA^®^ ring) and ovulation testing. The primary outcome variables measured pre- and post-intervention included a health history questionnaire, sleep quality (Pittsburg Sleep Quality Index, PSQI) dietary intake (Diet History Questionnaire III, DHQ III), body composition (dual energy X-ray absorptiometry, DXA; and bioelectrical impedance analysis, InBody S10), and muscular strength (TONAL^®^). Figure 1 shows the study experimental design and methodology.

### 2.3. Supplementation

CR (Creapure^®^ AlzChem, Trostberg, Germany; GRAS Notice No. GRN 931) and PLA (Globe^®^ Plus 10 DE Maltodextrin, Calgary, AB, Canada) were consumed Monday through Friday (1200–1400 h) in the Human Performance Laboratory at the University of Idaho for 6 consecutive weeks. At the Friday laboratory visit, participants were given two doses of their assigned supplement in a travel container to take home for consumption (1200 and 1400 h) on Saturday and Sunday and were instructed to return the container at the subsequent Monday laboratory visit. Photo and video evidence of supplement consumption was captured by participants and sent electronically to researchers for compliance assurance on weekend days. The CR powder (5 g·day) was mixed with an equal volume of maltodextrin (5 g corn-starch maltodextrin) and the PLA powder contained only maltodextrin (10 g corn-starch maltodextrin). Both products were similar in color (white), texture, and appearance. To optimize taste, participants were offered a choice of Crystal Light^®^ (Kraft Heinz, Mendota Heights, MN, USA) pink lemonade, raspberry lemonade, or fruit punch powder to add with 0.25 L of water for supplementation consumption. The purity of Creapure^®^ was established at >99.9% by independent laboratory testing (The Cary Company, Addison, IL, USA). Participant supplementation compliance of >80% was required for inclusion in data analyses (i.e., >34 out of 42 days).

### 2.4. Resistance Training Program

Participants completed a 6-week TONAL^®^ resistance training program (TONAL^®^ Corporation, San Francisco, CA, USA). TONAL^®^ is a commercial electromagnetic resistance wall-mounted device that all participants used 2x/week in the laboratory for their scheduled resistance training sessions. Training days were two non-consecutive days to allow for recovery. If participants missed a session, they had the opportunity to reschedule for a different time during the same week. Participants were required to maintain 80% adherence to continue participating in the study. Prescribed TONAL^®^ full body workouts ranged from 35–47 min where the first four sessions were ‘beginner’, and the remaining eight sessions were ‘intermediate’. All sessions were supervised by a trained researcher to ensure participant safety. TONAL^®^ technology increased or decreased load based on the parameters of the custom prescribed workout and participants received feedback on form during each movement.

### 2.5. Daily Testing

Throughout the 6-week intervention, daily sleep quantity and quality and ovulation testing were assessed. Additionally, daily testing included an adverse event ‘check-in’ to give participants the opportunity to report perceived issues related to the supplement and/or training. Participants visited the laboratory between 1200 and 1500 h to assess these measures, Monday through Friday, and measured the same outcomes at home between 1200 and 1500 h on Saturday and Sunday.

#### 2.5.1. Daily Sleep Assessment

Participants were given an ŌURA ring (ŌURA Health Oy, Oulu, Finland) for the duration of the 6-week study. The ŌURA ring assesses total sleep quantity and sleep quality, differentiating sleep into four distinct stages, awake, light sleep, deep sleep, and rapid eye movement (REM) sleep. ŌURA has been shown to be a valid device and comparable to gold-standard polysomnography [21].

#### 2.5.2. Ovulation Urine Testing

Participants were provided with a dixie cup and gloves to collect a urine sample. The urine sample was handed to a researcher who then dipped the ovulation test strip (Easy@Home Fertility, Premom, Burr Ridge, IL, USA) for 5–10 s until the dye moved up the stick. The stick was placed on a white, flat, non-absorbent surface for 5 min. The test results were recorded immediately after 5 min. A positive ovulation test was recorded if two lines were present. If a positive ovulation test occurred, the participant completed a bioelectrical impedance analysis (BIA; InBody S10, Gangam-gu, Seoul, Republic of Korea) to assess total body water (TBW; kg), intracellular water (ICW; kg), and extracellular water (ECW; kg) at the time of ovulation. Body composition methodology details are below in Section 2.6.3.

### 2.6. Pre- and Post-Testing

Participants visited the laboratory for a series of pre- and post-testing before and after the 6-week intervention. Pre- and post-testing laboratory visits occurred between 1200 and 1500 h and included health questionnaires, dietary assessment, body composition measures, and muscular strength testing which are all described in detail below.

#### 2.6.1. Health Questionnaires

Participants completed a health history questionnaire to assess general health history and menstrual cycle history. Questions included perceived experiences and symptoms before, during, and after menses. Then, participants completed the PSQI, which is a validated metric to assess self-reported sleep quality and fatigue. The PSQI asked participants to rate daily fatigue based on their perception of how it affected their day on a scale of 0–3 (0 = no problems with fatigue and 3 = extreme problem with fatigue).

#### 2.6.2. Dietary Intake Assessment

Participants completed the DHQ III (https://epi.grants.cancer.gov/dhq3/; URL accessed on 15 December 2022), pre- and post-intervention. DHQ III is an online food frequency questionnaire for adults of at least 19 years of age and is widely utilized by researchers to assess food and dietary supplement intake using 135 food and beverage line-items and 26 dietary supplement questions. Each line-item asked participants the frequency of food item intake and portion size in the previous 30 days. Total calories (kilocalories), carbohydrate (g and g/kg), fat (g), protein (g and g/kg), and estimated habitual CR (g) were used for analysis. To estimate CR content in foods, the following categories were used from DHQ III reports and were summed to calculate habitual CR intake [22]: meat (0.13 g/oz: beef, pork, veal, lamb, game), cured meat (0.06 g/oz: sausages and luncheon meat), organ meat (0.08 g/oz: liver and kidney), poultry (0.11 g/oz: chicken and turkey), and seafood (0.08–0.16 g/oz: salmon, herring, anchovies, trout, cod, tuna).

#### 2.6.3. Body Composition

Participants performed BIA (InBody S10, Gangam-gu, Seoul, Republic of Korea) at pre-intervention, ovulation, and post-intervention and dual energy X-ray absorptiometry (DXA; Hologic DXA Scanner, Hologic Horizon^TM^; Danbury, CT, USA) at pre- and post-intervention. Prior to BIA testing, participant hydration status was confirmed using a urine sample and refractometer (Atago 3749-E04, Tokyo, Japan). Participants were required to be euhydrated (1.002–1.025) to complete body composition testing. Participants first removed their socks and had their wrists and ankles wiped clean prior to electrode placement. The minimal detectable difference using InBody for TBW (kg) was 0.19 [23]. Total body water (TBW; kg), intracellular water (ICW; kg), and extracellular water (ECW; kg) were used for analysis. Next, one non-invasive anteroposterior view of the total body lying supine was performed using DXA. Testing was completed according to the manufacturer’s instructions and specifications. Results were analyzed with APEX software, version 4.5.2.1 (Hologic Inc., Marlborough, MA, USA). The quality analysis for the densitometer was conducted daily using a standard aluminum spine block (Hologic Phantom) provided by the manufacturer. Measurements of the phantom fell within the manufacturer’s precision standard with a coefficient of variation <0.5%. Test–retest interclass coefficient of variation (CV; %) using DXA for lean soft tissue (LST; kg) and fat mass (FM; kg) was 1.1% and 0.69%, respectively [23]. The minimum detectable differences using DXA for LST (kg) and FM (kg) were 0.27 and 0.21, respectively. LST (kg and %), FM (kg and %), visceral adipose tissue (VAT; cm^2^), and appendicular lean soft tissue index (ALSTI; kg/m^2^) were used for analysis. Body composition index (BCI) was calculated as follows: BCI = [(LSTpost − LSTpre) + (FMpre − FMpost)], where pre is pre-intervention and post is post-intervention, to evaluate overall body composition changes.

#### 2.6.4. Muscular Strength

All participants were familiarized with the TONAL^®^ equipment and then completed the TONAL^®^ Strength Assessment per the manufacturer’s specifications to generate a pre-and post-intervention total body TONAL^®^ strength score. The pre-intervention TONAL^®^ Strength Assessment established initial weight/intensity for TONAL^®^ training sessions based on a 3-repetition maximum lift of four primary movements including bench press, neutral grip deadlift, seated lat pulldown, and seated shoulder press. Participants completed this assessment using prompts from the TONAL^®^ program and safety monitoring from a trained researcher.

### 2.7. Statistical Analyses

All statistical analyses were conducted using SPSS version 29 (IBM Corp., Armonk, NY, USA) and jamovi version 2.2.5 [24] with statistical significance set at 0.05. ANOVAs were conducted to ensure that no significant group (CR vs. PLA) differences in baseline measures were observed. Repeated-measures ANOVAs were used to examine whether there were group (CR vs. PLA) differences across time (pre, post). Any significant interactions were decomposed using Bonferroni-corrected post hoc tests.

Given that participants provided daily measures across a six-week intervention, a multilevel modeling (MLM) approach was used to assess changes in sleep duration (total sleep, deep sleep, REM sleep, light sleep) measured via the ŌURA ring. The MLM approach was used not only to account for the nested structure of the data (i.e., repeated measurements within subjects) but also because of its ability to include participants with missing data in the analyses [25]. Using the GAMLj module in jamovi, group (CR = 1; PLA = 0)*week (1, 2, 3, 4, 5, 6) MLMs were conducted to examine whether sleep duration changed across the six weeks by group status. A random effect of participant was included in these models, while week was only included as a fixed effect since the models failed to converge when including it as a random effect. To examine whether sleep duration changed by group status following a workout, additional group (CR = 1; PLA = 0) x workout (non-workout = 0; workout = 1) MLMs were conducted, including random effects of participant and workout. All MLMs used the restricted maximum likelihood estimation method, the Satterthwaite method to approximate degrees of freedom, and the Wald method to construct 95% CIs. Any significant main effects or interactions were decomposed by using Bonferroni-corrected post hoc tests.

## 3. Results

### 3.1. Sleep

#### 3.1.1. Sleep Quality

There were no significant differences in the PSQI between groups or from pre- to post-intervention. Notably, the majority of CR and PLA groups had a Global PSQI score < 5, indicating poor sleep quality at both pre-testing (55%, 66%, respectively) and post-testing (55%, 75%, respectively).

#### 3.1.2. Sleep Duration

There were no significant group differences in sleep duration (total sleep, deep sleep, REM sleep, light sleep) or changes by week across the intervention for deep and REM sleep duration. Across both groups, there were significant differences by week in total sleep (F(5,87.16) = 2.33, *p* = 0.049) and light sleep duration (F(5,88.19) = 2.38, *p* = 0.045); however, none of the post hoc tests was significant (total sleep duration: Week 1: 427.4 min, SE = 11.66; Week 2: 438.1 min, SE = 11.97; Week 3: 413.0 min, SE = 11.66; Week 4: 412.8 min, SE = 11.97; Week 5: 399.1 min, SE = 12.12; Week 6: 428.6 min, SE = 12.29; light sleep duration: Week 1: 235.1 min, SE = 11.11; Week 2: 248.4 min, SE = 11.31; Week 3: 229.0 min, SE = 11.11; Week 4: 221.6 min, SE = 11.31; Week 5: 222.5 min, SE = 11.40; Week 6: 240.6 min, SE = 11.50).

On workout and non-workout days (one participant’s data were excluded given that they only had less than 10% complete data; therefore, these analyses included data from the remaining 20 participants), there was a significant group x workout interaction for total sleep duration (F(1,85.80) = 4.15, *p* = 0.045). Post hoc tests (Bonferroni-corrected alpha = 0.05/2 = 0.025) indicated no group differences (PLA: 422.26 min, SE = 13.41; CR: 431.83 min, SE = 11.06) in total sleep on non-workout days (t(17.89) = 0.55, *p* = 0.588); however, compared to the PLA group (M = 396.70 min, SE = 13.89), the CR group (M = 444.60 min, SE = 11.62) experienced an increase in their total sleep duration on workout days (t(30.92) = 2.65, *p* = 0.013; Figure 2). There were no other group differences or effects by workout for deep sleep, REM sleep, and light sleep duration.

### 3.2. Participant Demographics

Of the 27 participants enrolled, 2 withdrew prior to beginning the study and 4 withdrew at various points during the study following pre-testing. Data reflect participants who completed both pre- and post-testing and were >80% compliant regarding supplementation and exercise training. Participant demographics at baseline are displayed in Table 1. There were no significant differences between groups.

### 3.3. Strength

TONAL^®^ strength score improved from pre- to post-intervention for both groups (F(1,19) = 125.57, *p* < 0.001); however, there were no group differences or group × time interaction in strength score.

### 3.4. Body Composition

There were no significant group differences in DXA body composition metrics; however, both groups significantly increased absolute FM but not relative fat mass (kg; F(1,19) = 8.83, *p* = 0.008), TBW (F(2,28) = 5.09, *p* = 0.013), ICW (F(2,28) = 5.49, *p* = 0.010), and ECW (F(2,28) = 3.49, *p* = 0.041; Table 2). Although there were significant differences in TBW, ICW, and ECW from pre-intervention to ovulation testing, there were no differences in these metrics from ovulation to post-intervention.

### 3.5. Dietary Intake Assessment

There were no significant group differences in dietary behaviors or changes from pre- to post-intervention. Across both groups, there was a significant reduction in the following from pre- to post-intervention: total calories (*p* = 0.039), protein (*p* = 0.018), and fat (*p* = 0.036). Relative protein (*p* = 0.009; g/kg) and carbohydrate (*p* = 0.023; g/kg) significantly decreased from pre- to post-intervention (Table 3).

## 4. Discussion

Unique and important results of this study included greater sleep duration (quantity) the night following resistance training exercise bouts with CR supplementation compared to resistance training exercise without CR. These results are consistent with previous research among sleep-deprived participants, which has shown restorative effects of CR supplementation with and without exercise on cognitive performance and mood improvement [7,26]. However, these results contradict those identified in a rat model, in which total sleep duration decreased following 4 weeks of CR supplementation [27]. These results have not been reproduced in a human model and warrant further investigation. Current evidence suggests that CR supplementation may have greater effects on the brain under stressful conditions such as exercise and sleep deprivation and that CR may decrease the effects of sleep deprivation [28]. In the present study, it is possible that resistance training provided a great enough stimulus to stress the body and therefore enhance the effects of CR supplementation. While the precise mechanism for this change is unknown, CR has been shown to decrease oxidative stress and inflammation and improve brain bioenergetics, which may have favorable effects on sleep [28]. Furthermore, these findings are important given that young females typically experience greater sleep disturbances and poorer sleep quality compared to males [11,12].

Although there is a dearth of CR supplementation and sleep research, several studies have investigated the impact of resistance training on sleep [29,30,31,32,33,34]. Findings vary, with one study reporting no effects of acute resistance training on sleep assessed via polysomnography [31]. It is possible that lab-based sleep assessments fail to capture real-life sleep among participants when free-living conditions (e.g., the habitual sleep environment) are removed. At-home sleep assessment using ŌURA^®^ or similar daily sleep-tracking devices may better capture participants’ sleep responses to resistance training exercise. Other research findings suggest subjective sleep quality improves following 12–16-week resistance training programs; however these findings have been observed among clinical and older male and female populations [32,34]. Moreover, many studies report subjective sleep quality only using the PSQI before and after exercise intervention. In the present study, there were no differences in perceived sleep quality using the PSQI over time or between groups, indicating a need for future research to include both sleep quality and quantity measures in dietary and exercise interventions that are short in duration (e.g., 6–10 weeks).

In the present study, both groups significantly improved TONAL^®^ strength score, while no significant changes in relative LST and FM were identified. It is likely that the improvements in TONAL^®^ strength scores are attributed to improved neural function [35]. Our findings do not suggest that CR supported these improvements during the short 6-week intervention as all participants improved over time. It is possible that the training period was too short to elicit significant changes in LST and FM, and it may require a training program longer in duration and higher intensity to yield changes previously demonstrated in other female-specific exercise intervention studies [36,37]. The small samples size likely limited the ability to detect small changes between CR and PLA. A previously reported perceived mild side effect of CR supplementation is bloating [38], which may be more prominent among women. Participants did not report any perceived negative side effects with CR supplementation, and no significant changes in TBW, ICW, and ECW between groups suggest that cellular fluid did not change as a result of CR supplementation. However, significant changes in TBW, ICW, and ECW were observed between pre-testing and ovulation, indicating cellular fluids may change specifically during ovulation. These results are limited and warrant follow-up in future investigations given the small number (n = 16) of participants reporting a positive ovulation test during the present 6-week intervention.

Estimated habitual CR consumption was low for non-vegetarians/vegans where participants consumed less than 0.5 g/day at pre- and post-testing, when 2–4 g would be expected from a habitual omnivore diet [1]. This could partly be attributed to low total estimated energy intake below 2000 calories at both pre- and post-testing. Interestingly, all participants experienced significant decreases in total calories consumed, relative protein and carbohydrate intake, and absolute fat intake. The reasons for these changes among all participants are unclear. The food frequency questionnaire used in this study (DHQ III) captured the previous 30 days, so these changes represent the difference in diet behavior 30 days prior to the study and 30 days during the study. It is possible that the significant decreases in relative protein below the recommended dietary allowance (<0.8 g/kg/day) also impacted participants’ ability to stimulate muscle protein synthesis, thus impacting our body composition findings. Research has indicated that 0.8 g/kg/day is insufficient at stimulating muscle protein synthesis for those engaging in resistance training, and at least 1.2 g/kg/day protein should be consumed to stimulate muscle protein synthesis [39]. To mitigate significant dietary changes in future research, it may be beneficial to incorporate a nutrition education session prior to an intervention to maximize potential benefits of additional CR.

Key strengths of the present study include the use of an at-home sleep-monitoring device to assess sleep quantity throughout the 6-week intervention in addition to sleep quality. The study employed double-blinding and randomized, placebo-controlled methods where participants reported to the laboratory daily and submitted weekend videos to ensure supplement compliance. The study also included naturally menstruating women; a population that has historically been excluded from research. Despite these strengths, there are a few limitations to address. Given that total saturation of CR stores takes approximately 28 days [40] it is possible that a 6-week intervention was too short to produce significant changes in LST and may impact our sleep-related findings. Future sleep and CR research should employ CR loading phases and experiment with greater CR dosages that may have greater effects on the brain and sleep [28]. The resistance training intervention utilized new and innovative resistance training equipment, TONAL^®^. The study relied on TONAL^®^ programming and strength testing and it is possible that the intensity of the programming was insufficient to produce LST changes compared to standard resistance training. Greater than two days per week of resistance exercise would have likely yielded difference results, however, the study was designed to standardize exercise across all participants to observe the additive effects of creatine supplementation.

Future research should explore the effects of CR supplementation, both alone and in combination with varying types of exercise, on cognitive performance, mood, and its relationship to sleep quantity using daily measurement devices such as ŌURA^®^. Due to individual menstrual differences and the fact that ovulation was not detected for some participants we were unable to determine a precise menstrual cycle phase for all participants in our study. There is a strong need to include women in research, and the inability to determine menstrual cycle phase should not necessarily deter researchers from including female participants. Additionally, studies should investigate a similar intervention using females on hormonal birth control to explore the potential interaction of hormonal birth control and CR on sleep following exercise.

## 5. Conclusions

This study demonstrated that CR supplementation may increase acute total sleep when combined with resistance training among naturally menstruating women. These findings may be particularly important for those consuming a diet habitually low in creatine sources even if they follow an omnivore diet. While much research has investigated the impact of resistance training on sleep, the additive effect of resistance training and creatine supplementation on sleep quality and quantity is further warranted.

## Figures and Tables

**Figure 1 nutrients-16-02772-f001:**
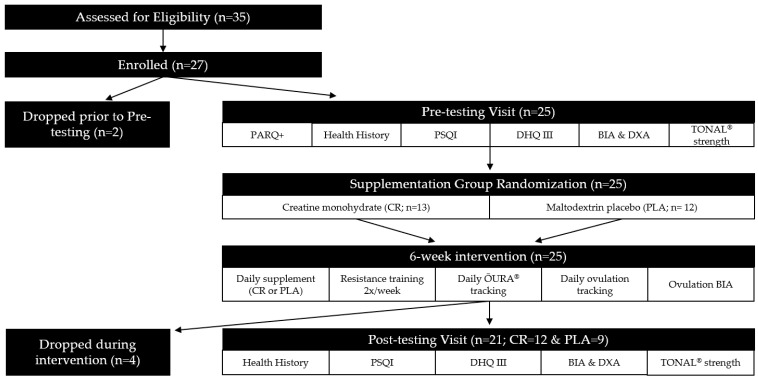
Experimental Design & Methodology Schematic. PARQ+, Physical Activity Readiness Questionnaire for Everyone; PSQI, Pittsburgh Sleep Quality Index; DHQ III, Diet History Questionnaire III; BIA, bioelectrical impedance analysis; DXA, dual energy x-ray absorptiometry; CR, creatine; PLA, placebo.

**Figure 2 nutrients-16-02772-f002:**
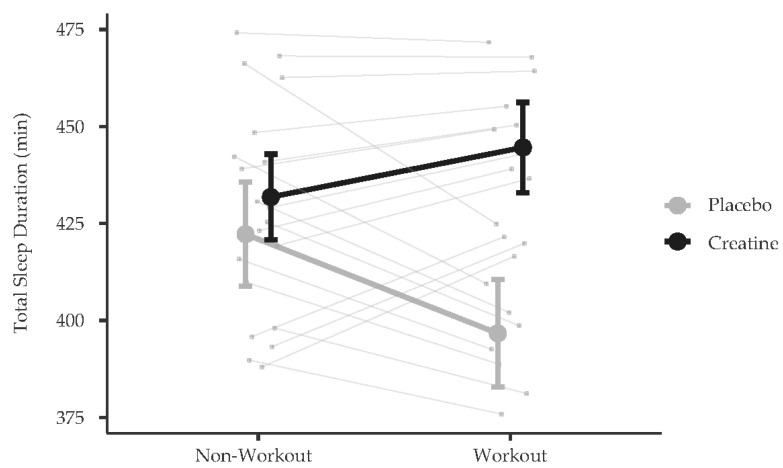
Model-estimated means (and standard errors) of total sleep duration as a function of workout and group status. Note: The thick lines represent the average effect estimated by the fixed effect and the thinner lines represent the effects by participant (i.e., the random effect).

**Table 1 nutrients-16-02772-t001:** Participant demographics at baseline.

	All (n = 21)	Placebo (n = 9)	Creatine (n = 12)
Age	20 ± 2	21 ± 2	20 ± 2
Height (cm)	164.2 ± 5.4	165.8 ± 4.1	161.9 ± 6.3
Body mass (kg)	66.8 ± 14.0	68.6 ± 15.6	65.4 ± 13.3
BMI (kg/m^2^)	24.8 ± 5.3	23.9 ± 3.8	25.9 ± 6.8
Menstrual cycle phase	F: 62%	F: 56%	F: 66%
	L: 14%	L: 11%	L: 17%

Data reported as mean ± SD. cm, centimeters; kg, kilograms; m, meters; F, follicular; L, luteal. Note: Not all participants had a confirmed menstrual cycle phase at baseline.

**Table 2 nutrients-16-02772-t002:** Body Composition Metrics.

	Placebo			Creatine		
	Pre (n = 9)	Post (n = 9)	Ovulation (n = 4)	Pre (n = 12)	Post (n = 12)	Ovulation (n = 12)
DXA Measures						
BM (kg)	68.6 ± 15.6	69.6 ± 15.3 ^^^	-----	65.4 ± 13.3	66.8 ± 13.0 ^^^	-----
FM (%)	31.9 ± 4.6	32.61 ± 5.1	-----	34.8 ± 6.1	35.2 ± 6.1	-----
FM (kg)	22.3 ± 7.7	23.2 ± 8.0 ^^^	-----	23.4 ± 8.6	24.1 ± 8.6 ^^^	-----
ST (%)	63.8 ± 4.3	63.9 ± 4.7	-----	62.1 ± 6.4	61.5 ± 5.6	-----
LST (kg)	43.9 ± 7.9	43.7 ± 7.7	-----	39.9 ± 5.1	40.5 ± 5.3	-----
VAT (cm^2^)	56.5 ± 29.8	65.5 ± 33.0	-----	102.3 ± 68.7	91.7 ± 46.2	-----
ALSTI (kg/m^2^)	6.9 ± 1.0	6.8 ± 0.9	-----	6.3 ± 1.0	6.4 ± 0.9	-----
BCI (kg)	-----	−0.86 ± 1.7	-----	-----	−0.17 ± 2.3	-----
BIA Measures						
TBW (kg)	32.5 ± 4.2	33.7 ± 4.6 ^^^	33.4 ± 3.1 ^#^	33.6 ± 3.6	34.1 ± 2.9 ^^^	34.7 ± 2.7 ^#^
ICW (kg)	20.3 ± 2.6	21.2 ± 2.8 ^^^	21.1 ± 2.1 ^#^	21.1 ± 2.3	21.5 ± 1.8 ^^^	21.9 ± 1.8 ^#^
ECW (kg)	12.1 ± 1.6	12.5 ± 1.7 ^^^	12.4 ± 1.2 ^#^	12.5 ± 1.3	12.6 ± 1.1 ^^^	12.8 ± 0.9 ^#^

Data reported as mean ± SD. BM, body mass; FM, fat mass; LST, lean soft tissue; VAT, visceral adipose tissue; ALSTI, appendicular lean soft tissue index; BCI, body composition index; TBW, total body water; ICW, intracellular water; ECW, extracellular water; kg, kilograms; cm, centimeters; m, meters. ^^^ Indicates statistical significance from pre to post *p* < 0.05. ^#^ Indicates statistical significance from pre to ovulation *p* < 0.05.

**Table 3 nutrients-16-02772-t003:** Dietary Intake.

	Placebo		Creatine	
	Pre (n = 9)	Post (n = 9)	Pre (n = 12)	Post (n = 12)
Calories (kcal)	1826 ± 383	1561 ± 698 ^^^	1929 ± 1020	1394 ± 608 ^^^
Protein (g)	64 ± 26	50 ± 24 ^^^	70 ± 28	51 ± 20 ^^^
Protein (g/kg)	1.0 ± 0.3	0.7 ± 0.3 ^^^	1.0 ± 0.6	0.8 ± 0.4 ^^^
Carbohydrate (g)	241 ± 54	191 ± 90	217 ± 133	160 ± 78
Carbohydrate (g/kg)	3.7 ± 1.2	2.6 ± 1.0 ^^^	3.5 ± 2.5	2.6 ± 1.5 ^^^
Fat (g)	64 ± 24	55 ± 30 ^^^	73 ± 30	53 ± 22 ^^^
Habitual Creatine (g)	0.39 ± 0.26	0.25 ± 0.11	0.35 ± 0.25	0.26 ± 0.19

Data reported as mean ± SD. kcal, kilocalories; g, grams; kg, kilograms; ^^^ Indicates statistical significance from pre to post *p* < 0.05.

## Data Availability

The data sets generated in the present study are available from the corresponding author upon reasonable request.

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
