# Peer review of "Creatine Improves Total Sleep Duration Following Resistance Training Days versus Non-Resistance Training Days among Naturally Menstruating Females"

_nutrients, 2024, doi:10.3390/nu16162772_

Round 1

Reviewer 1 Report

Comments and Suggestions for Authors

Dear authors.

Congrats for the research titled “Creatine Improves Total Sleep Duration Following Resistance Training Days Versus Non-Resistance Training Days Among Naturally Menstruating Females”. It is an interesting research; however, it is necessary to take into account some recommendations so that the manuscript can be improved:

Abstract:

Provides information on the various parts of the manuscript.

Introduction:

Was information collected about the phases of the sample's menstrual cycle? It would be convenient to provide some information in this regard. If it was not done, it is recommended to introduce it in future lines of research.

Materials and Methods:

Complete. Provide adequate information to be able to replicate the experiment.

Results:

Well organized and structured. Tables and figures are appropriate

Discussion:

In the results section (L 244-247) it said that “3.1.2. Sleep duration. There were no significant group differences in sleep duration (total sleep, deep sleep, REM sleep, light sleep) or changes by week across the intervention for deep and REM sleep duration…” However, in the first paragraph of the Discussion, it is stated that “Unique and important results of the present study was that the combination of CR supplementation and resistance training resulted in greater total sleep duration.” Further explanation is required based on the results

Conclusions:

After the justification of the previous comment, the conclusions may need to be revised

References:

Ok.

With the application of these changes, the quality of the manuscript will be improved.

Thank you

Author Response

Thank you for the comments. All edited text is included in the manuscript in red. 

Comment 1: Was information collected about the phases of the sample's menstrual cycle? It would be convenient to provide some information in this regard. If it was not done, it is recommended to introduce it in future lines of research.

Thank you for the comment. Our study captured a full menstrual cycle for all participants and therefore data was collected at each point of the menstrual cycle. We have added information into Table 1 (Participant Demographics) reflecting the percentage of participants in luteal and follicular phases at baseline. We have also added a sentence in 2.2 Experimental Design clarifying what information was collected from participants. However, not all participants ovulated, making it impossible to determine precise menstrual cycle phases for all participants. This has been added into our future directions. The revised text now reads:

Pg 2: “Menstrual cycle phase was noted at the beginning of the intervention.” 

Pg 7, Table 1: Participant demographics at baseline.

Pg 10: “Due to individual menstrual differences and the fact that ovulation was not detected for some participants we were unable to determine a precise menstrual cycle phase for all participants in our study. There is a strong need to include women in research, and the inability to determine menstrual cycle phase should not necessarily deter researchers from including female participants.”

Comment 2: In the results section (L 244-247) it said that “3.1.2. Sleep duration. There were no significant group differences in sleep duration (total sleep, deep sleep, REM sleep, light sleep) or changes by week across the intervention for deep and REM sleep duration…” However, in the first paragraph of the Discussion, it is stated that “Unique and important results of the present study was that the combination of CR supplementation and resistance training resulted in greater total sleep duration.” Further explanation is required based on the results

Reply: We apologize for any confusion we caused in the original submission. We re-structured the first paragraph of the Discussion to clarify that the primary finding relates to the effects on sleep duration following resistance training exercise bouts in the CR supplementation compared to placebo group. This revised paragraph now reads:

"Unique and important results of this study included greater sleep duration (quantity) the night following resistance training exercise bouts with CR supplementation compared to resistance training exercise without CR. These results are consistent with previous research among sleep-deprived participants, which has shown restorative effects of CR supplementation with and without exercise on cognitive performance and mood improvement [6,25]. Furthermore, these findings are important given that young females 

Reviewer 2 Report

Comments and Suggestions for Authors

Thank you for submitting your manuscript. It was a pleasure to read. We appreciate your efforts in this highly interesting field of research. 

We only have minor issues to be considered:

1. The manuscript would greatly benefit from a timescale that clarifies when participants started to supplement in comparison to training: Did training and supplementation start at the same time? 

Since we would assume filling CR stores takes some time, you should include this when discussing your results. 

2. Would you consider the observed effects an immediate reaction to the supplementation? 

3. Have you considered that intracellular CR stores first had to fill up to reach their maximum effect? 

Besides: We know there has not yet extensive research conducted regarding sleep and creatine supplementation. In fact, only one publication (to our knowledge) has dealt with CR and showed a decrease in sleep amount. There has, however, been some papers dealing with CR-supplementation and sleep deprivation. 

Therefore, we would suggest to include these projects in your discussion and adapt the latter accordingly: 

Creatine-supplementation reduces sleep need and homeostatic sleep pressure in rats (2017):

Reduces sleep pressure and total sleep time

Beyond muscle: the effects of creatine supplementation on brain creatine, cognitive processing, and traumatic brain injury (2018):

Effects of sleep deprivation in CR supplementation is reduced. 

Muscle creatine loading in men (1996):

Saturation of CR stores takes about 28 days

Author Response

Thank you for the comments. Please find our responses below. All edits in the manuscript are included in red font. 

Comment 1. The manuscript would greatly benefit from a timescale that clarifies when participants started to supplement in comparison to training: Did training and supplementation start at the same time? 

Since we would assume filling CR stores takes some time, you should include this when discussing your results. 

Thank you for the comment. This information provides necessary context to our study. In our study, training and supplementation began at the same time. We have added this sentence into our methods and discussed these items in our limitations. The revised text now reads:

Pg 2: “Once randomized, participants began supplementation and resistance training concurrently (Figure 1).”

Comment 2. Would you consider the observed effects an immediate reaction to the supplementation? 

Thank you for the question. This would likely not be an immediate reaction to creatine as supplemental creatine would need to enter circulation, then cross the blood-brain barrier and subsequently accumulate in brain tissue. Thus, it’s likely that our results are due to the accumulation of creatine over time leading to small improvements in sleep duration following exercise. While no mechanism was measured in this study, creatine has been shown to decrease oxidative stress, inflammation and improve brain bioenergetics which may have favorable effects on sleep.

Comment 3. Have you considered that intracellular CR stores first had to fill up to reach their maximum effect? 

Thank you for the comment. It is likely that intracellular CR stores had to fill up first and with the amount of time that it takes for CR stores to be fully saturated, it is likely that this did not happen until week 4 of the 6-week intervention. We have added a sentence into our limitations to address the need to a CR loading phase to fully saturate CR stores and the need for future studies to employ a greater CR dosage. Greater CR dosages have been associated with changes in cognitive performance, and therefor may be necessary for significant changes in sleep. The revised text now reads:

“Given that total saturation of CR stores takes approximately 28 days [40] it is possible that a 6-week intervention was too short to produce significant changes in LST and may impact our sleep-related findings. Future sleep and CR research should employ CR loading phases and experiment with greater CR dosages that may have greater effects on the brain and sleep [27].”

Besides: We know there has not yet extensive research conducted regarding sleep and creatine supplementation. In fact, only one publication (to our knowledge) has dealt with CR and showed a decrease in sleep amount. There has, however, been some papers dealing with CR-supplementation and sleep deprivation. 

Therefore, we would suggest to include these projects in your discussion and adapt the latter accordingly: 

Creatine-supplementation reduces sleep need and homeostatic sleep pressure in rats (2017):

Reduces sleep pressure and total sleep time

Thank you for the suggestion of adding this article. We have added this article into our discussion referencing the contradicting results between our study and the above study conducted in rats. The text now reads:

Pg 9: “However, these results contradict those identified in a rat model, in which total sleep duration decreased following 4-weeks of CR supplementation [26]. These results have not been reproduced in a human model and warrants further investigation.”

Beyond muscle: the effects of creatine supplementation on brain creatine, cognitive processing, and traumatic brain injury (2018):

Effects of sleep deprivation in CR supplementation is reduced. 

Thank you for the suggestion of adding this article. We have added this article into our discussion and limitations referencing the findings that CR supplementation may have an increased effect under stressful conditions including exercise and sleep deprivation. The revised text now reads:

Pg 9: “Current evidence suggests that CR supplementation may have greater effects on the brain under stressful conditions such as exercise and sleep deprivation and that CR may de-crease the effects of sleep deprivation [27]. In the present study, it is possible that resistance training provided a great enough stimulus to stress the body, and therefor enhance the effects of CR supplementation, however the mechanism in which CR increased total sleep is unknown [27].”

Pg 10: “Future sleep and CR research should employ CR loading phases and experiment with greater CR dosages that may have greater effects on the brain and sleep”

Muscle creatine loading in men (1996):

Saturation of CR stores takes about 28 days

Thank you for the suggestion of adding this article. We have added this article into our limitations. The revised text now reads”

Pg 10: “Given that total saturation of CR stores takes approximately 28 days [40] it is possible that a 6-week intervention was too short to produce significant changes in LST and may impact our sleep-related findings”